# Metallic nanocrystals with low angle grain boundary for controllable plastic reversibility

Qi Zhu[1,7], Qishan Huang[2,7], Cao Guang[1,7], Xianghai An[3], Scott X. Mao [4], Wei Yang[2], Ze Zhang[1], Huajian Gao [5,6], Haofei Zhou [2✉] & Jiangwei Wang [1✉]

Advanced nanodevices require reliable nanocomponents where mechanically-induced irreversible structural damage should be largely prevented. However, a practical methodology to improve the plastic reversibility of nanosized metals remains challenging. Here, we propose a grain boundary (GB) engineering protocol to realize controllable plastic reversibility in metallic nanocrystals. Both in situ nanomechanical testing and atomistic simulations demonstrate that custom-designed low-angle GBs with controlled misorientation can endow metallic bicrystals with endurable cyclic deformability via GB migration. Such fully reversible plasticity is predominantly governed by the conservative motion of Shockley partial dislocation pairs, which fundamentally suppress damage accumulation and preserve the structural stability. This reversible deformation is retained in a broad class of face-centred cubic metals with low stacking fault energies when tuning the GB structure, external geometry and loading conditions over a wide range. These findings shed light on practical advances in promoting cyclic deformability of metallic nanomaterials.

[1] Center of Electron Microscopy and State Key Laboratory of Silicon Materials, School of Materials Science and Engineering, Zhejiang University, Hangzhou 310027, China. [2] Center for X-Mechanics, Department of Engineering Mechanics, Zhejiang University, Hangzhou 310027, China. [3] School of Aerospace, Mechanical and Mechatronic Engineering, The University of Sydney, Sydney, NSW 2006, Australia. [4] Department of Mechanical Engineering and Materials Science, University of Pittsburgh, Pittsburgh, PA 15261, USA. [5] School of Mechanical and Aerospace Engineering, College of Engineering, Nanyang Technological University, Singapore 639798, Singapore. [6] Institute of High Performance Computing, A*STAR, Singapore 138632, Singapore. [7]These authors contributed equally: Qi Zhu, Qishan Huang, Cao Guang. ✉email: haofei_zhou@zju.edu.cn; jiangwei_wang@zju.edu.cn

Nanoscale materials are widely anticipated to be used as building blocks for advanced wearable devices[1,2], flexible electronics[3,4] and micro/nanoelectromechanical systems (MEMS/NEMS)[5,6] due to their unparallel physical and mechanical properties. Both experimental and theoretical studies in the past two decades have revealed a wealth of unique mechanical responses, such as size-dependent strengthening[7], ultrahigh strength[8], superplasticity[9] and anelasticity[10]. However, much less research has been focused on the deformation reversibility in nanosized materials, which is of general significance to the functionality and reliability of integrated nanocomponents in flexible/wearable devices. In contrast to the elasticity-dominated responses of semiconductor nanomaterials[11], metallic nanomaterials commonly experience non-conservative defect activities associated with frequent heterogeneous surface nucleation[12]. The resultant irreversible shear localization and structural degradation[13] notoriously compromise their service reliability[14,15]. Hence, the structural stability and damage resistance of metallic nanocomponents need to be precisely engineered to fundamentally retard the cumulative degradation[16]. Nevertheless, the development of a practical methodology to realize plastic reversibility in nanoscale metals without damage accumulation remains a challenge.

Although several approaches for realizing recoverable plasticity have already been demonstrated in metallic nanomaterials, including twinning/phase transformation-induced lattice reorientation[17–19] and penta-twin-dominated dislocation retraction[20], these deformation mechanisms are highly orientation- and material-dependent. Moreover, asymmetrical twinning/detwinning shear[21] and the associated huge surface kinks[19] inevitably lead to deformation instability and irreversible structural damage in metallic nanocrystals, which compromises their long-time plastic deformability. Additional investigations suggest that grain boundary (GB)-mediated deformation may contribute to the recovery of plastic strain in nanostructured metals[22–24]. Likewise, in bulk polycrystalline metals, high densities of internal interfaces, such as GBs[25] and nanoscale twin boundaries (TBs)[26], have been widely adopted to alleviate damage accumulation in loading cycles. In particular, low angle GBs (LAGBs) allowing slip transmission endow bulk metals with enhanced crack resistance[27].

Inspired by the success of interface engineering in bulk materials[26,27], here, we propose an approach of GB design in metallic nanocrystals to achieve controllable plastic reversibility. Through integrated state-of-the-art in situ nanomechanical testing and molecular dynamics (MD) simulation, we demonstrate that face-centred cubic (FCC) metallic nanocrystals with custom-designed LAGBs can accommodate exceptional reversible plasticity with negligible damage accumulation. The extraordinary plastic reversibility is governed by the collective motion of dissociated GB dislocations that readily overtake any non-conservative defect activities. This energetically favoured conservative GB migration leads to the reversible deformation of metallic nanocrystals with a variety of intrinsic GB structures and external geometries under different loading conditions. These findings hold implications for interface engineering of metallic nanomaterials towards controllable reversible deformability, further boosting the optimal design of reliable nanocomponents from the bottom up.

## Results

**Reversible migration of LAGB in shear cycles.** Prior to the nanomechanical testing, Au bicrystals with controlled GB misorientations were fabricated by in situ welding of two nanoscale single crystals inside a transmission electron microscope (TEM,

see "Methods" for details). Figure 1a shows an example of an as-prepared Au bicrystal with a diameter of 16.2 nm and a misorientation ($\theta$) of 13.5° between the lattices of the top and bottom grains. A [1$\bar{1}$0] tilt LAGB (delineated by the yellow dotted line in Fig. 1a) is located close to the top of the nanocrystal. Closer examination shows that the GB consists of orderly aligned dislocations with a Burgers vector of 1/2 [01$\bar{1}$] (Fig. 1j), which accommodate the misorientation between the adjoining crystals. The average distance between these GB dislocations is measured to be 1.05 nm (roughly 3–5 times of the interatomic distance), consistent with the theoretical prediction based on the linear relation between the reciprocal of dislocation spacing ($D$) and GB misorientation ($\theta \sim b/D$)[28]. These GB dislocations are aligned in a staggered manner, indicating stress relaxation of the as-fabricated GB in the Au bicrystal.

Reversible shear loading was horizontally applied to the Au bicrystal (with the top grain fixed to the bulk, see the inset in Fig. 1k) to explore the deformation behaviour of the Au bicrystal. With leftward shear, the LAGB gradually migrated downward (Fig. 1b). After a migration distance of ~10 nm, the LAGB reached the bottom of the bicrystal (Fig. 1c). Subsequently, reversed shear loading was imposed, activating upward migration of the LAGB (Fig. 1d). After a full cycle of shear loading, the LAGB completely returned to its initial position (Fig. 1e). Notably, the external geometry of the deformed bicrystal is nearly identical to its original structure, suggesting a fully reversible deformation of the Au nanocrystal with the LAGB. We noted an inconspicuous change in the atomic configuration of the LAGB, which was probably induced by the nonuniform distribution of shear stress near the GB. To further validate the migration reversibility of the LAGB (Fig. 1f–h), we subjected the same nanocrystal to additional shear loading cycles. After five cycles, the original structure of the nanocrystal was well retained (Supplementary Movie 1), except for a slight surface variation due to the localized surface diffusion (see the differences between the surface configurations in Fig. 1a, i)[29]. In each cycle, the migration of the LAGB was conservative (i.e., without defect nucleation or annihilation) and fully reversible, indicating good structural stability upon cycling. Such a unique deformation mechanism contributed to a maximum reversible shear strain of 0.25 (i.e., the lateral displacement of the bottom grain divided by the uniform gauge length of the bicrystal). The migration distances of the LAGB in each cycle are plotted as a function of the shear displacement to quantify the migration reversibility (Fig. 1k). Apparently, all migration loops of the LAGB show identical GB migration-shear displacement coupling, which rationalizes the exceptional structural stability throughout the loading cycles. The average shear coupling factor $\beta$ (defined as the ratio between the shear displacement and the migration distance of the LAGB) was measured as 0.27, consistent with the value estimated based purely on the geometry of the LAGB ($\beta \sim \theta$).

**Atomistic mechanism of reversible GB migration.** The dynamic evolution of the 13.5° [1$\bar{1}$0] LAGB was further characterized to uncover the detailed migration mechanism upon fully reversed shear loading (Fig. 2). Atomistic observations show that the reversible migration of the LAGB was dominated by the collective movement of dissociated GB dislocations throughout the shear loading cycles (Fig. 2a–c). Neither lattice dislocation nucleation nor defect annihilation associated with the GB or free surface was captured in our experiments, in stark contrast to the extensive dislocation activities commonly observed in the deformation of metallic single crystals with similar sizes[13,19]. Before loading, the orderly aligned GB dislocations showed negligible dissociation, as reflected by the corresponding geometrical phase analysis (GPA,

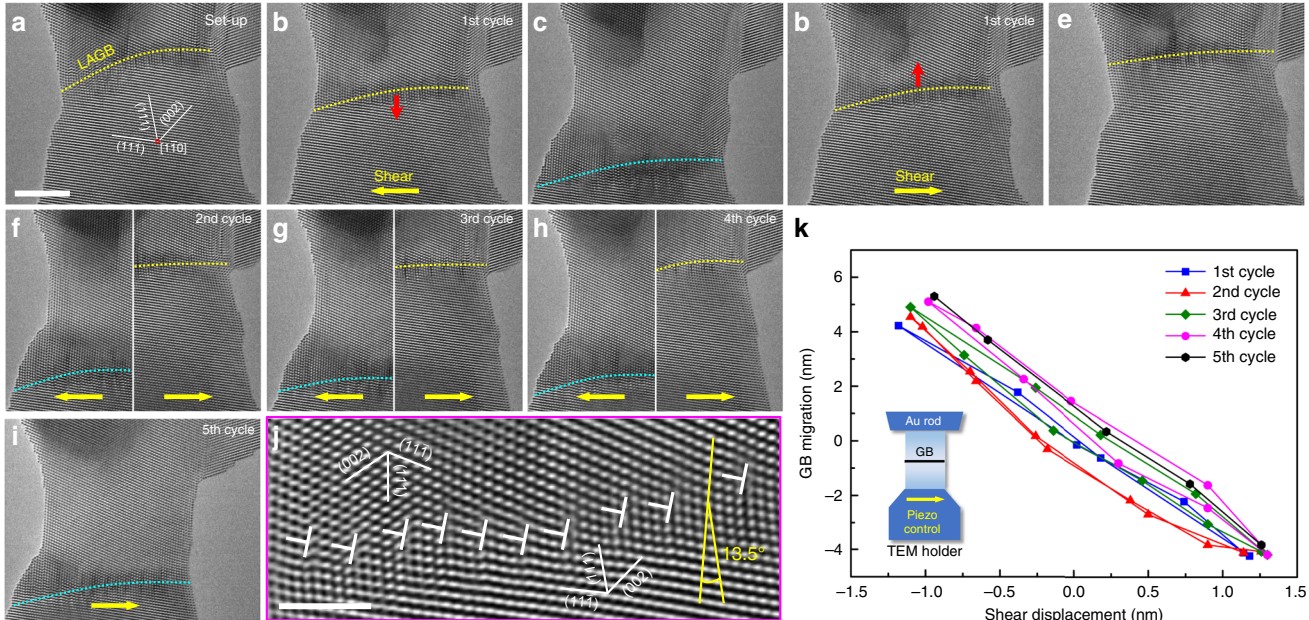

**Fig. 1 Reversible migration of a 13.5° $[1\bar{1}0]$ low angle grain boundary (LAGB) in shear loading cycles. a** An as-fabricated gold (Au) bicrystal with an LAGB. **b**, **c** Downward migration of the LAGB under leftward shear loading. **d**, **e** Upward migration of the LAGB under rightward shear loading. The directions of shear loading and GB migration in each cycle are shown by the yellow and red arrows, respectively. **f–h** Reversible migration of this LAGB in subsequent shear loading cycles. For clear demonstration, the top and bottom positions of the LAGB in each cycle are delineated by the yellow and light blue dotted lines, respectively. **i** The LAGB after the left shear in the fifth cycle. **j** Atomistic structure of the 13.5° $[1\bar{1}0]$ LAGB, which consists of aligned dislocations with a Burgers vector of $1/2\,[01\bar{1}]$. **k** GB migration distance versus shear displacement of the bottom grain in the first five loading cycles. Scale bars: **a** 5 nm and **j** 2 nm.

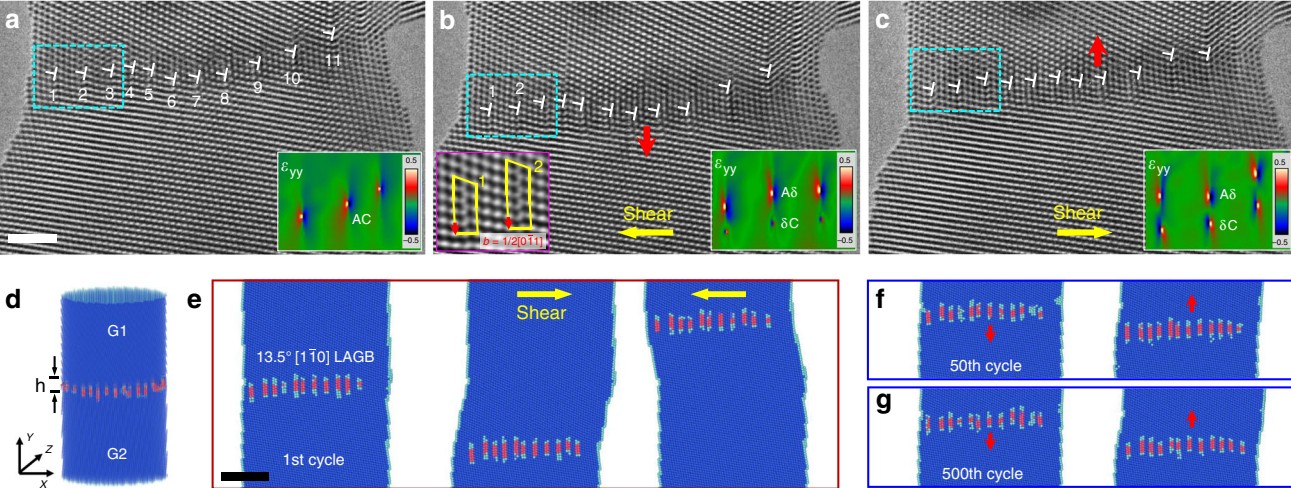

**Fig. 2 Dissociation of the 13.5° $[1\bar{1}0]$ LAGB in fully reversed loading cycles. a–c** Zoomed-in deformation snapshots showing the atomistic structure of the migrating 13.5° $[1\bar{1}0]$ LAGB in reversed shear loading. **a** Initial LAGB, consisting of well-aligned $1/2\,[01\bar{1}]$ dislocations with negligible dissociation. **b**, **c** GB migration associated with evident dissociations under leftward (**b**) and rightward (**c**) shear loading. The insets in **a–c** exhibit geometrical phase analysis (GPA) maps of the vertical normal strain ($\varepsilon_{yy}$), demonstrating the dissociation of GB dislocations 1–3, as marked by the light blue rectangles. The zoomed-in TEM image in **b** shows the atomistic core structure of dissociated GB dislocations 1 and 2. **d** Perspective structure of a fully relaxed Au bicrystal with a dissociated 13.5° $[1\bar{1}0]$ LAGB (indicated by the arrows) constructed in molecular dynamics (MD) simulation at 300 K. The diameter and length of the bicrystal are 15 and 30 nm, respectively. **e** MD simulation snapshots demonstrating the reversible migration of the 13.5° $[1\bar{1}0]$ LAGB in a shear loading cycle. **f**, **g** Reversible deformation of the Au bicrystal with the 13.5° $[1\bar{1}0]$ LAGB in long-time shear loading cycles. The MD snapshots present well-retained LAGB-mediated reversible deformation in the **f** 50th and **g** 500th shear cycles, respectively. The cyan and red atoms indicate the Shockley partial dislocations and the stacking fault ribbons, respectively. Scale bars: **a** 2 nm and **e** 5 nm.

see inset of Fig. 2a). Upon deformation, the GB dislocations tended to dissociate into pairs of Shockley partial dislocations, (i.e., **Aδ + δC**) on the coherent (111) slip planes of the top and bottom grains (Fig. 2b and Supplementary Fig. 1), as confirmed by the extended atomic configurations in the high resolution

TEM image and evident strain dipoles in the corresponding GPA map (see insets). The dissociated nature of the GB greatly facilitated the continual and correlated gliding of GB dislocations inside the bicrystal, contributing to the fully reversible migration of this LAGB (Fig. 2c).

MD simulations of a cylindrical bicrystal sample with a 13.5° $[1\bar{1}0]$ LAGB can further rationalize the underlying GB dislocation dynamics that govern the reversible deformation of the Au bicrystal during shear loading cycles (Fig. 2d and Supplementary Figs. 2, 3). The size of the simulated sample is the same as that of the real bicrystal in the experiment. The simulation results confirm the instant dissociation of pre-existing GB dislocations upon loading (Fig. 2e), suggesting that the dissociated GB configuration is energetically favourable and relatively stable during the GB motion. Upon the reversible deformation, these Shockley partial dislocation pairs (with a stacking fault ribbon, Supplementary Fig. 4) glided smoothly in both grains and promptly reversed their trajectory upon switching the shear direction, consistent with the experimental observation. It is evident that the number of GB dislocations in the LAGB remained constant throughout the loading cycles, confirming the conservative deformation behaviour of the LAGB. This phenomenon may arise from the geometrical requirement of GB dislocations to maintain the misorientation and the large resolved normal stress (on the slip planes) that impedes diffusive climb of GB dislocations. With these GB dislocations as pre-existing plastic carriers, non-conservative deformation induced by massive lattice dislocation activities (including the nucleation, motion and annihilation) was greatly suppressed. Owing to the slightly non-uniform distribution of the localized shear stress inside the nanocrystal, the GB dislocations may not necessarily dissociate to the same extent (see dislocations 1 and 2 in Fig. 2b). Such unique deformation behaviour of the LAGB endows the metallic bicrystal with exceptional plastic reversibility yet fundamentally-absent damage accumulation (Fig. 2e and Supplementary Movies 2 and 3). To verify this deformation paradigm on a long-term basis, the Au bicrystal was further deformed for more than 500 shear cycles in our MD simulation (Fig. 2f, g and Supplementary Movie 4). Both the structural stability of the nanocrystal and the migration reversibility of the LAGB were well retained, indicating excellent plastic reversibility of this Au bicrystal with a characteristic LAGB.

**Generality of reversible dynamics among different GBs**. Both our experiments and simulations clearly demonstrate that the bicrystal with LAGB exhibits impressive plastic reversibility via the highly organized motion of dissociated GB dislocations. Figure 3a schematically illustrates the dissociation of an LAGB in the bicrystal, where two orientation parameters, i.e., the tilt GB misorientation ($\theta$) and the inclination between the GB plane and shear direction ($\alpha$), are elucidated. The dissociated GB configuration preserves the slip continuity between the lattices of the top and bottom grains[28] and enables simultaneous gliding of GB dislocations in both grains. This configuration, to some extent, eliminates the necessity of additional lattice defects as deformation carriers and thereby greatly facilitates the stable reversible deformation of the nanocrystal. Given that dissociation of GB dislocations is a common phenomenon for LAGBs, especially in metals with low stacking fault energies[30], the reversible deformation behaviour is expected to be general in bicrystals with different LAGBs. To substantiate the commonality of this reversible deformation mechanism of LAGBs, additional in situ experiments were carried out by varying the misorientation of the <110> tilt GB in the range of 8°–19° (Supplementary Figs. 5–7), and the GB-mediated reversible deformation were observed up to 8 shear loading cycles (Supplementary Fig. 7). MD simulations demonstrate that similar plastic reversibility prevails in Au bicrystals with GB misorientations in the range of 8.80°–25.06° (Supplementary Fig. 8), validating the generality of our findings.

Referring to the significance of the GB misorientations, the GB energy and the dissociation width of GB dislocations were quantitatively analysed by MD simulations (Fig. 3b). The GB dissociation widths generally decrease with increasing GB misorientation, with an evident plateau corresponding to the relaxed state of GB dislocations in the range of 12°–20° (Supplementary Discussion 1). Below and above this misorientation range, the GB configurations could be categorized into unconfined and restricted stages, respectively (elucidated by the inset GB structures). In contrast, the GB energy increased nearly monotonically and steadily with increasing GB misorientation, which arises from a growing density of GB dislocations and their interactions (quantified in Supplementary Fig. 19). When the misorientation exceeded 24°, the dislocation-type GB could not stably exist, and non-conservative defect nucleation from either the GB or free surface dominated the plastic deformation in shear loading cycles (Supplementary Fig. 9). Consequently, shear localization and damage accumulation occurred in the Au bicrystals, compromising the plastic reversibility. Additional MD simulations of a variety of FCC metallic bicrystals (Supplementary Fig. 10) show similar LAGB-mediated reversible deformability and its misorientation dependence (Fig. 3c). Interestingly, the upper limit of misorientation for GB dissociation lies in a narrow range of 24°–28° among different metals (indicated in Fig. 3c), suggesting that the dissociation is controlled predominantly by the GB geometry. Consistently, the experimental evidence shows that no observable GB dissociation occurred in Au bicrystals with misorientations larger than 22° (Supplementary Fig. 11). More importantly, this misorientation dependence of GB dissociation exerts a direct influence on the GB mobility. As shown in Fig. 3d, the GB migration rate decreased monotonically with the increasing misorientation ($\theta$) under a constant shear velocity, leading to an increase of the shear coupling factor ($\beta$).

Quantitative simulations revealed that the mechanical responses of the Au bicrystals also showed a strong GB misorientation dependence (Fig. 3e). During parallel reversible shear loadings with a maximum strain amplitude of 0.5, a dynamic asymmetry between the applied shear stresses $\sigma_{max}$ and $\sigma_{min}$ (inducing the downward and upward GB migrations, respectively) was unambiguously demonstrated with respect to the GB misorientation. The resolved shear stress for slip in the top grain increased monotonically with the misorientation (beyond 10°), which kinetically favoured the upward migration of the GB, as reflected by a continuously reducing $\sigma_{min}$. In contrast, the resolved shear stress for dislocation slip in the bottom grain remained unchanged (due to the simulation setup), whereas a higher slip discontinuity was intrinsically associated with the GBs with larger misorientations, which impeded the downward GB migration, resulting in an increasing $\sigma_{max}$. At $\theta \sim 19°$, $\sigma_{max} = |\sigma_{min}|$, which corresponds to a symmetrical orientation of the coherent (111) slip planes in the adjacent grains with respect to the GB. Nevertheless, this shear loading asymmetry (characterized as normalized shear stress, $R = |\sigma_{max}/\sigma_{min}|$) does not perfectly coincide with the theoretically predicted misorientation dependence of the normalized Schmid factor (i.e., the reciprocal ratio of the Schmid factor for slip in the bottom and top grains, denoted as $\gamma = |m_{AC,min}/m_{AC,max}|$), as illustrated in Fig. 3f. The discrepancy between the normalized shear stress and normalized Schmid factor may originate from the characteristic dissociation of the GB, which introduces significant lattice distortion near the GB and thus compromises the validity of the theoretical predictions based on a homogeneous lattice. The normalized shear stress gradually approaches the trend of the normalized Schmid factor, consistent with the decreased GB dissociation tendency at larger misorientations (Fig. 3b).

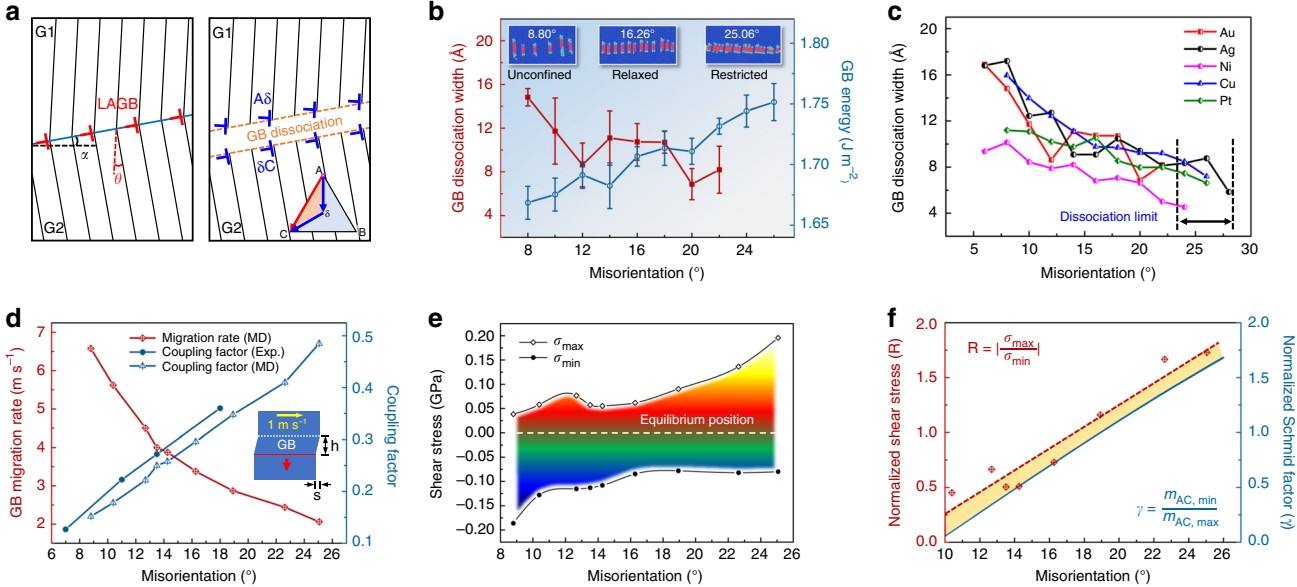

**Fig. 3 Generality of the reversible deformation in a variety of metals with different dislocation-type GBs. a** Geometrical model of the LAGB configuration before (left) and after (right) dissociation. The GB misorientation and inclination are indicated by θ and α in the schematic model, respectively. The inset illustrates the GB dissociation into two Shockley partials on the (111) slip plane. **b** GB dissociation width and corresponding GB energy for different misorientations. Error bars represent the standard deviations from statistical analyses, where $n$ ranges from 6 to 13 for GB dissociation width (depending on the GB misorientation) and $n = 3$ for GB energy. **c** Variations in the GB dissociation width with increasing GB misorientation in different face-centred cubic (FCC) metallic bicrystals. The dashed lines mark the upper limits of the misorientation range for stable GB dissociation in different FCC nanocrystals. **d** Statistics of migration rates and shear coupling factors ($\beta = s/h$, illustrated by the inset schematic) of LAGBs with different misorientations. **e** Ranges of applied shear stresses with respect to the GB misorientation. $\sigma_{max}$ and $\sigma_{min}$ denote the shear stresses that induced GB migration towards the bottom and top positions in each cycle, and the interval stress states are differentiated by the gradient colours. **f** Normalized applied shear stresses (R = $|\sigma_{max}/\sigma_{min}|$, red diamonds) derived from MD simulations, and normalized Schmid factors ($\gamma = m_{AC,min}/m_{AC,max}$, blue curve) predicted from the geometrical model. The red dashed line is a linear fitting of the normalized shear stresses.

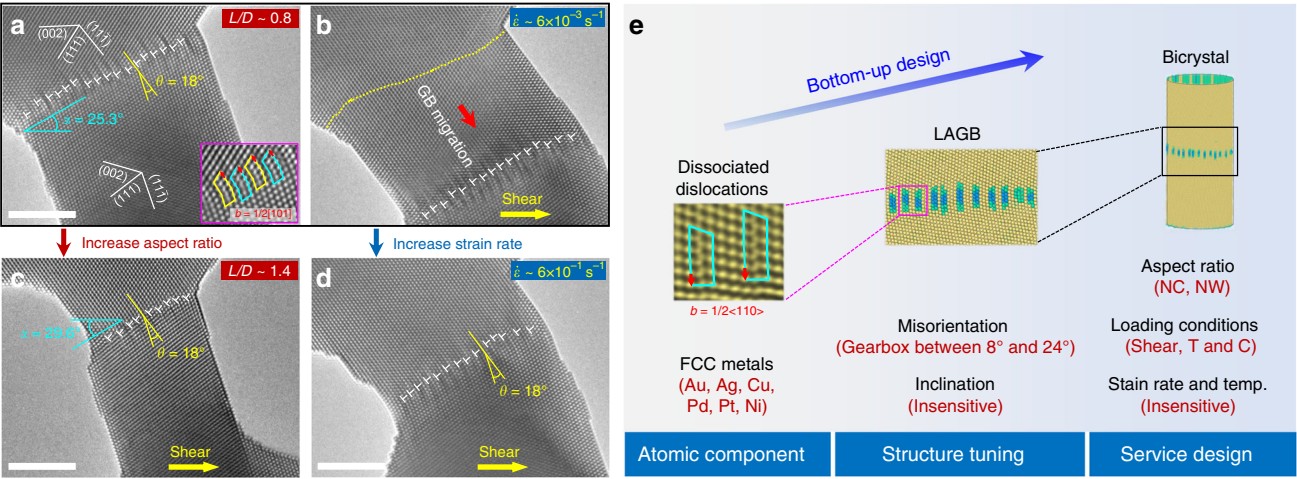

**Fig. 4 Bottom-up design of plastic reversibility in Au bicrystals with dislocation-type GB. a** Au bicrystal consisting of an 18° [1$\bar{1}$0] GB, with an inclination (α) of 25.3°. The inset shows the staggered configuration of the dissociated GB dislocations (**b** = 1/2[101]), as marked by the yellow and light blue concave rectangles. **b** GB-mediated shear deformation of the Au bicrystal under a strain rate of $6 \times 10^{-3}\,s^{-1}$. **c** Deformation snapshot of the reversible deformation of another Au bicrystal with almost twice the aspect ratio (L/D) and near-identical θ and α compared with **a**. **d** GB-mediated reversible deformation of an Au bicrystal with the same 18° [1$\bar{1}$0] GB as **a** under a higher strain rate of $6 \times 10^{-1}\,s^{-1}$. **e** Schematic illustrating the bottom-up design protocol of metallic bicrystals to realize superior plastic reversibility. NC and NW represent nanocrystal and nanowire; T and C stands for uniaxial tension and compression; temp. is an abbreviate for temperature. Scale bars: 5 nm.

**Factors influencing the plastic reversibility**. Both the experimental and simulation studies underpin a robust plastic reversibility during the cycling of metallic bicrystals with different LAGBs. To rationalize this unique deformation paradigm, other potential influencing factors, including the GB inclination, nanocrystal diameter, aspect ratio and deformation strain rate,

were systematically investigated. Figure 4a shows an additional Au bicrystal consisting of an 18° [1$\bar{1}$0] dislocation-type GB with an average inclination (α) of 25.3°. Although the GB misorientation and inclination of this nanocrystal were larger than those of the 13.5° [1$\bar{1}$0] GB (Fig. 2), reversible GB migration still dominated the deformation in shear loading cycles. A shear strain

as high as 0.36 was perfectly accommodated by the collective motion of dissociated GB dislocations (Fig. 4b) without inducing observable lattice bending or grain rotation (Supplementary Fig. 12). The large GB inclination was proved to have negligible influence on the reversible deformability of dislocation-type GB, which probably originated from the invariable resolved shear stresses on GB dislocations when adjusting the GB inclination (see Fig. 3a). This potent nature of GB-mediated deformation essentially maintains the plastic reversibility in differently oriented metallic nanocrystals with various diameters, as manifested by both experiments (Supplementary Fig. 13) and simulations (Supplementary Fig. 14a, b). Figure 4c and Supplementary Fig. 14c further indicate that the reversible migration of dislocation-type GBs can be well retained for a wide range of aspect ratios ($L/D$) (summarized in Supplementary Tables 1 and 2). Notably, for similar GB misorientation and inclination (compared with Fig. 4a), doubling the aspect ratio of the Au nanocrystal (from ~0.8 to ~1.4) led to no fundamental difference in the plastic reversibility. Additionally, reversible plasticity can be realized in Au bicrystals when tuning the aspect ratio ($L/D$) between 2 and 6 (despite the lower mobility under the larger aspect ratio), while GB motion could hardly be detected for $L/D > 6$ (Supplementary Fig. 14c). Theoretical analysis suggests that the applied shear stress inevitably induces a large bending moment on the metallic nanocrystals with an excessively large aspect ratio, impeding the correlated motion of GB dislocations (see Supplementary Discussion 2). In addition, an increasing number of surface flaws in large-aspect-ratio nanocrystals[31] could impair the inherently conservative deformations. Moreover, the applied shear strain rate was found to have negligible influence on the reversible migration of dislocation-type GBs (over at least two orders of magnitudes) under the strain rates of $10^{-3}$–$10^{-1}$ s$^{-1}$ in the experiments (Fig. 4d and Supplementary Movie 5) and $10^{7}$–$10^{9}$ s$^{-1}$ in the MD simulations (Supplementary Table 2), indicating the rate-insensitive plastic reversibility. Complementary MD simulations further demonstrated that the plastic reversibility of Au bicrystals ($L/D = 2$) is even preserved in the presence of surface roughness and intragranular vacancies (Supplementary Fig. 15a, b). In view of the fact that LAGB migration ($\theta = 16°$) also proceeded via conservative dislocation motion upon tension (Supplementary Fig. 16), we can reasonably expect a plastic recovery of nanocrystals under reversed uniaxial loading. It was indeed confirmed by our MD simulations that LAGB-mediated reversible deformation dominated in metallic bicrystals under uniaxial tension-compression cycling (Supplementary Fig. 15c). Besides, an Au bicrystal nanowire with a large aspect ratio of 10 exhibited a similar reversible deformation behaviour via LAGB migration under uniaxial tension and unloading (Supplementary Fig. 15d), indicating a broad range of validity for the reported phenomena.

## Discussion

To engineer nanoscale structures with high tolerance against cyclic damage, we can either stay in the elastic regime[11], or develop special materials that enable reversible plastic deformation. Phase transformation[17] and twinning-detwinning[18] are two common methods of realizing plastic reversibility in nanoscale metals, which, however, are often limited by severe orientation-dependence and lack of controllability. Here, we demonstrate a mechanism of GB-mediated plastic reversibility that enables reversible deformation of nanosized materials beyond their simple elastic limit, and further validate it under the influences of multiple governing factors, as summarized in Supplementary Tables 1 and 2. Such stable plastic reversibility via shear coupled GB migration (over 500 shear loading cycles in MD simulations,

Fig. 2g) holds over a wide range of FCC metals (Au, Ag, Cu, Pd, Pt, Ni) when systematically tuning the GB structures (misorientation angle ranging from 8° to 24°, inclination angle), nanocrystal geometries (aspect ratio, surface roughness) and loading conditions (shear, tension, compression, strain rate, temperature), etc. As schematically illustrated in Fig. 4e, the universally dissociated GB dislocations in different FCC metals provide principal building blocks for the reversible deformation, which enables us to design metallic nanocrystals with controlled reversible deformability and high damage tolerance. Thus, this GB approach with targeted or optimal design can push the limit of controlled plastic reversibility in metallic nanomaterials[32,33].

Theoretically, this superior plastic reversibility of metallic bicrystals mainly originates from conservative motion of the readily dissociated GB dislocations that fundamentally suppress the irreversible damage accumulation arising from the pronounced heterogeneous surface nucleation and annihilation of partials or twins. Intrinsically, the orderly aligned dissociated GB dislocations (i.e., Shockley partial dislocation pairs bound with a stacking fault ribbon) accommodate the lattice misorientation between neighbouring grains and reduce the Peierls barrier[28], enabling the smooth migration of LAGB with negligible frictional heat. As shown in Fig. 3e, the maximum applied shear stress of ~0.2 GPa is lower than the estimated critical resolved shear stress (0.47 GPa) for surface nucleation of a partial dislocation in Au nanocrystals[13]. Therefore, the activation of LAGB migration is energetically favoured over heterogeneous defect nucleation from the free surface, leading to the dominance of GB migration in the reversible deformation of metallic bicrystals. The absence of non-conservative defect nucleation and annihilation helps preclude shear localization[34] and GB structure changes[35], and the reversible deformation can be well-retained, causing negligible damage accumulation in the nanocrystal. However, when the misorientation exceeds the upper limit, the very closely spaced dislocation cores prohibit the motion of dislocations, resulting in localized shearing[36]. Therefore, the GBs fail to migrate continuously and even become increasingly disordered (Supplementary Fig. 9a–c). The consequent non-conservative nucleation of lattice defects (including dislocations and stacking faults)[37] from these GBs can further deteriorate the synergistic gliding of GB dislocations, impairing the inherent reversible deformability upon shear cycling (Supplementary Fig. 9d–g). Based on both experimental and simulation results, the transformational misorientations of a disordered GB are in the range of 26–28° among different FCC metals. The dislocation character of GBs can stably exist at almost twice the typical misorientation of 15° defined by the classic description of LAGBs[28], which can be ascribed to the fact that the structure of high angle GB can transit from the structure unit type to the dislocation type (similar to the classic LAGB) with the decreasing sizes[38]. Nevertheless, the highly organized GB motion is viable as long as the dislocations nature within the GB is retained. Most significantly, in contrast to many high angle GBs consisting of heterogeneous structure units[37], dislocation-type GBs can easily accommodate reversible deformation, given that the as-formed Shockley partial dislocation pairs serve as correlated and smooth carriers of plasticity in both grains. However, below the lower misorientation limit of ~8°, the few GB dislocations can no longer accommodate large shear strain of the bicrystal nanowire and surface nucleation is in turn activated (Supplementary Fig. 8a), thus compromising the reversible deformability. Besides, the well-preserved surface tomography and stable geometry of the bicrystals are two extrinsic governing factors of good plastic reversibility, which precludes irreversible damage accumulation from shear localization[19] and necking[13] by eliminating heterogeneous defect nucleation. Further investigations suggest that this GB-mediated

plastic reversibility is comparatively insensitive to surface imperfections, such as surface steps or terraces (Figs. 1, 4 and Supplementary Fig. 15), where non-uniform stress distributions often occur. As a result, the synergy of all above factors, including the dissociated GB structure, energetically favoured conservative GB migration and well-preserved nanocrystal geometry, allows for stable plastic reversibility without tangible damage accumulation, endowing the metallic nanocrystals with potentially exceptional cyclic deformability.

In conclusion, we have proposed a GB engineering approach to realize reversible deformability in metallic nanocrystals over a wide range of set-up orientations and loading conditions. The well-documented dislocation-type GBs are the engineering elements that could be implanted into nanocrystals where the misorientation of these custom-designed GBs serve as the tool to control the amplitude and strain rate of reversible deformation. Notably, the nanocrystal geometry (e.g., aspect ratio) should be coupled with the GB structure for the optimal design of metallic bicrystals (Fig. 4e). A maximum shear strain of $\sim\beta(\theta)$ can be stably retained under long-time loading cycles, based on both experimental and simulation results (Figs. 1, 2). We further demonstrate that such GB-dominated reversible deformation mechanism can be generally applicable to FCC metallic nanocrystals (Fig. 3c and Supplementary Fig. 10) with low stacking fault energies. As an outlook, similar plastic reversibility is likely to retain for other types of GBs, e.g., <100> tilt LAGBs, given the characteristic GB structure of 1/2 <110> dislocation arrays[39]. Moreover, this reversible deformation mechanism can be extended into multi-grain systems, where the migration of upper and bottom GBs is generally consistent with that in bicrystals (Supplementary Fig. 17). From the nanotechnology perspective, the realization of custom-designed GB can benefit from the widely adopted controlled epitaxial growth technique[40,41], by which high-quality metallic bicrystals with continuous variation in boundary orientation can be readily fabricated, e.g., 90° <110> and 48° <111> tilt GBs. We also need to note that nanocrystalline metals typically possess much more complex GB structures and the plastic deformation is usually controlled by the coordinated deformation of numerous grains and the associated different types of GBs, which deserves systematic investigation in future work. These findings provide a feasible strategy to tune the cyclic deformability of metallic nanomaterials from the bottom up, which enables us to design reliable metallic nanocomponents for high-performance nanodevices. The precisely controlled plastic reversibility mediated by LAGBs in metallic nanocrystals also has certain implications for stable energy conversion/dissipation and mechanical damping in NEMS and flexible devices.

## Methods

**In situ TEM nanofabrication and nanomechanical testing.** In situ nanofabrication and fully reversible shear loading of Au bicrystals with different <110> GBs were conducted using a PicoFemto® TEM electrical holder from Zeptools Co. inside a FEI Titan Cs-corrected TEM. Prior to the shear loading, defect-free Au bicrystals were fabricated via an in situ welding technique. First, two bulk Au rods (99.99 wt.%, Alfa Aesar Inc.) with a diameter of 0.25 mm were cut by a ProsKit wire cutter to obtain clean fracture surfaces with numerous single crystalline nanoscale tips; then, the fractured Au rods were loaded onto the static and probe sides of the TEM electrical holder. Afterwards, the Au probe was actuated by a built-in piezo manipulator (behind the probe) so that it could approach the rod on the static side of the holder. Upon contact, the nanoscale tips on both sides were welded together in situ inside the TEM by pre-applying a voltage potential of approximately −2 V at the probe side. A defect-free Au bicrystal with a tilt GB was thus fabricated by taking advantage of the orientation differences between the single crystalline nanoscale tips on opposite sides. In this way, a broad class of GB structures can be fabricated by carefully selecting the orientations and sizes of the nanoscale single crystals on both sides.

During in situ shear loading experiments, the Au probe was precisely controlled to alternately move leftward/rightward to impose reversible shear loading at a constant velocity of $\sim0.005$ nm s$^{-1}$, which gave rise to an estimated strain rate at the level of $\sim10^{-3}$ s$^{-1}$. For each GB, a near-identical reversible migration amplitude was imposed among all shear cycles by carefully controlling the shear distance (i.e., lateral motion of the probe). In all experiments, the TEM was operated at 300 kV with low beam intensity to minimize the potential beam effects on the deformation mechanisms; in situ experiments were recorded by a Gatan 994 charge-coupled device (CCD) camera at a rate of $\sim0.3$ s per frame.

**Molecular dynamics simulations.** MD simulations were carried out on Au bicrystals with a total of ~31,1000 atoms using Large-scale Atomic/Molecular Massively Parallel Simulator (LAMMPS)[42] and the embedded atom method (EAM) potentials for Au[43]. A cylindrical bicrystal model with a diameter of 15 nm and a total height of 30 nm (15 nm height for each grain) was created by constructing two separate crystals with a crystallographic misorientation of between 8° and 30° and joining them along the axial direction. GBs with different misorientations were generated by tilting the top grain of the bicrystal around the <110> axis while fixing the bottom grain. Three boundary layers of atoms at the top and bottom of the system were fixed as rigid slabs. The remaining dynamic atoms were allowed to adjust their positions in a Nose-Hoover thermostat at 300 K. Free boundary conditions were applied in all three directions. The system was relaxed for 20 ps to obtain the equilibrated GB structure. The average GB energy under each misorientation was determined by calculating the energy difference between the relaxed bicrystal system and the two individual crystals before joining.

During the fully reversible deformation, a constant shear velocity of $v = 1$ m s$^{-1}$ parallel to the boundary plane was applied on the rigid slab of the top grain. Once the GB reached the pre-set position, the shear was reversed. The time step of the MD simulations was 2 fs. A velocity profile with a linear gradient from 0 to 1 m s$^{-1}$ was assigned to the dynamic atoms along the axial direction (Supplementary Fig. 1). The average vertical displacement of the GB atoms was recorded as the GB migration displacement. Ovito[44] was used to visualize the bicrystal model, and the common neighbour analysis was employed to identify the dissociation of the GBs during the simulations. Atoms with FCC, hexagonal close-packed (HCP) and disordered structures were marked in blue, red and cyan, respectively.

## Data availability

The data that support the findings of this study are available from the corresponding authors upon reasonable request.

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

## Acknowledgements
J.W. acknowledges the support of Basic Science Center Program for Multiphase Evolution in Hypergravity of the National Natural Science Foundation of China (51988101), the National Natural Science Foundation of China (51771172 and 51701179) and the Innovation Fund of the Zhejiang Kechuang New Materials Research Institute (ZKN-18-Z02). H.Z. acknowledges financial support from the National Natural Science Foundation of China (11902289) and computational support from the National Supercomputer Center in Tianjin and the Super Cloud Computing Center in Beijing. X.A. acknowledges support from the Australian Research Council under DE170100053 and from The University of Sydney under the Robinson Fellowship Scheme. Z.Z. acknowledges support from the National Natural Science Foundation of China (11234011 and 11327901).

## Author contributions
J.W. designed the experiments and directed the project. Q.Z., G.C. and J.W. conducted the experiments and analysed the data. Q.Z. and J.W. wrote the paper. Q.H. and H.Z. performed the simulations, analysed the data and revised the paper. X.A., S.M., W.Y., Z.Z. and H.G. contributed to the data analysis and the paper revision.

## Competing interests
The authors declare no competing interests.
