## [Peer Review File · Nature Communications]

Editorial note: This manuscript has been previously reviewed at another journal that is not operating a transparent peer review scheme. This document only contains reviewer comments and rebuttal letters for versions considered at *Nature Communications*.

Reviewers' Comments:

Reviewer #1:

Remarks to the Author:

Building on the three substantial reviewer comments, in comparison to the previous version, one notes that the authors responded lengthy to the various comments, but added few comments in order to fundamentally improve the manuscript.

While they clarified that this finding is only applicable to low stacking fault materials, they still fall short in providing more than rather vague arguments, such as epitaxial grown bicrystals or similar with very high angle boundaries (exceeding the working range from this work), towards an engineering solution in order to employ this very specific observation. Furthermore, the multigrain extension remains largely speculative.

Point-to-point Response to the Reviewer's comments

Manuscript ID: NCOMMS-20-12790A

Title: Metallic nanocrystals with low angle grain boundary for controllable plastic reversibility

Authors: Qi Zhu, Qishan Huang, Guang Cao, Xianghai An, Scott X. Mao, Wei Yang, Ze Zhang, Huajian Gao, Haofei Zhou, Jiangwei Wang

We sincerely thank the editor and the reviewer for their time and efforts in carefully reading our manuscript and their valuable comments and constructive suggestions on our work. In the following, the review comments are laid out in *italicized font* and our response to each comment is given in blue text. The manuscript has been revised accordingly.

Response to Reviewer #1

Building on the three substantial reviewer comments, in comparison to the previous version, one notes that the authors responded lengthy to the various comments, but added few comments in order to fundamentally improve the manuscript.

While they clarified that this finding is only applicable to low stacking fault materials, they still fall short in providing more than rather vague arguments, such as epitaxial grown bicrystals or similar with very high angle boundaries (exceeding the working range from this work), towards an engineering solution in order to employ this very specific observation. Furthermore, the multigrain extension remains largely speculative.

Response: We thank the reviewer for handling our revised manuscript. In fact, to fundamentally improve the manuscript, we have clearly elucidated our strategy to achieve flexible and controllable reversible plasticity, which is complemented with a systematic comparison with well-documented mechanisms in previous works. Moreover, we proposed a justifiable estimation of the large reversible shear strain in our work, discussed the orientation limit of this approach and provided additional experimental evidence for the potential application of LAGB-dominated reversible deformation in uniaxial loading cycles (other than merely shear loading cycles). Finally, we specified some techniques currently available in the nanotechnology community that shed light on realizing custom design of GB structures. Therefore, the plastic reversibility of metallic nanomaterials proposed in current study should possess potential engineering impact, although we are aware that further investigation and development are required. In our previous submission, we have thoroughly revised our manuscript and clarify all the questions raised by the three referees. Here, we further revised our manuscript to tone down some of the text about the engineering of these bicrystals in the final version, in response to the concerns of both Reviewer #1 and the editorial request.